# Effect of Silicon Nitride and Graphene Nanoplatelets on the Properties of Aluminum Metal Matrix Composites

**DOI:** 10.3390/ma14081898

**Published:** 2021-04-10

**Authors:** Rokaya Abdelatty, Adnan Khan, Moinuddin Yusuf, Abdullah Alashraf, Rana Abdul Shakoor

**Affiliations:** Center for Advanced Materials (CAM), Qatar University, 2713 Doha, Qatar; Rokaya.abdelatty@qu.edu.qa (R.A.); adnan.khan@qu.edu.qa (A.K.); moinuddin@qu.edu.qa (M.Y.); aa.ashraf@qu.edu.qa (A.A.)

**Keywords:** aluminum, composites, microwave sintering, microstructure, nanoindentation, hardness, strength

## Abstract

This research work aims at investigating the influence of a fixed content of silicon nitride (Si_3_N_4_) and varied contents of graphene nanoplatelets (GNPs) on the physical (density, structural, morphological) and mechanical properties (microhardness, nanoindentation) of Al-Si_3_N_4_-GNPs composites. The composites were fabricated by a microwave-assisted powder metallurgy route. The Si_3_N_4_ concentration was fixed at (5 wt.%) in Al-Si_3_N_4_-GNPs composites while the GNPs concentration was varied between (0 wt.%) to (1.5 wt.%) with an increment of (0.5 wt.%). The structural analysis indicates the formation of phase pure materials with high crystallinity. The microstructural analysis confirmed the presence of the Si_3_N_4_ and GNPs showing enhanced agglomeration with the increasing amount of GNPs. Moreover, the surface roughness of the synthesized composites increases with an increasing amount of GNPs reaching its maximum value (RMS = 65.32 nm) at 1.5 wt.% of GNPs. The Al-Si_3_N_4_-GNPs composites exhibit improved microhardness and promising load-indentation behavior during nanoindentation when compared to pure aluminum (Al). Moreover, Al-Si_3_N_4_-GNPs composites demonstrate higher values of compressive yield strength (CYS) and ultimate compressive strength (UCS) when compared to pure Al despite showing a declining trend with an increasing amount of GNPs in the matrix. Finally, a shear mode of fracture is prevalent in Al-Si_3_N_4_-GNPs composites under compression loading.

## 1. Introduction

Metal matrix composites (MMCs) are promising sophisticated materials that incorporate a metallic matrix such as aluminum (Al) and reinforcement particles which commonly are affiliated to families of borides, carbides, nitrides, and oxides [1,2]. In the family of MMCs, the aluminum metal matrix composites (AMMCs) have attracted much attention due to their distinctive properties such as low density, high strength, good wear resistance, and excellent thermal stability [3,4,5]. These features lead AMMCs to become a prospective contender for comprehensive utilization in the automobile and aviation industries [6]. At the same time, research have demonstrated that the incorporation of nano-sized reinforcements in AMMCs has exhibited superior behavior in enhancing their structural, thermal, and mechanical properties when compared with micron-sized reinforcements [7,8].

Nowadays, the production of Al-composites with nano-sized reinforcements undergoes multiple types of experimental research owing to the large specific surface area represented by the nanoparticles, which give rise to a strong contact interfacial area between the base material and its reinforcement [9,10,11]. This characteristic improves the mechanical behavior of the composites resulting in a superior nanotribological performance [12]. Different types of reinforcements such as metallic, amorphous, and ceramic particles, have been utilized in AMMCs for enhancing their performances. The selection of the reinforcements depends on the eventual desired and targeted properties of the AMMCs [13].

Recent studies have indicated that the properties are remarkably influenced by the nature of reinforcement, its weight fraction, and distribution within the metal matrix [6,14,15,16]. To optimize the properties of the AMMCs, the reinforcement material utilized is required to be uniformly distributed within the matrix. Different mixing techniques such as solution-assisted mixing, molten or liquid metal processes and ball milling has been applied to the reinforcements to obtain their uniform distribution within the metal matrix. Among these techniques, the ball milling process has attracted wide attention for its considerable impact on the microstructure of the product. Once utilized at the optimum operating conditions that conform with the feed material, it produces homogeneously distributed reinforcements within the metal matrix by enhancing the affinity between the reinforcement and the matrix [17].

Moreover, the ultimate properties of the AMMCs depend on the fabrication method. Several fabricating techniques have been reported for the manufacturing of AMMCs such as casting, thermomechanical processing, and the powder metallurgy (PM) route. The PM method is the most employed fabricating technique for the production of AMMCs owing to its ability in maintaining a high level of homogeneity of distributed reinforcements [18]. The PM method usually consists of four main stages. The first stage is the powder preparation stage where the powders of the aluminum matrix and the reinforcements are prepared. The second stage is the ball milling stage where the powders of aluminum and reinforcements are mixed. Powder products are then taken to the third stage of cold pressing to create cylindrical pellets where it will be taken for the final stage to microwave sintering to form the final homogenous product [19].

In the present study, silicon nitride (Si_3_N_4_) was selected as the first reinforcement based on its significant properties such as high hardness, compressive strength, wear-resistance, and high thermal stability [20,21]. Additionally, graphene nanoplatelets (GNPs) were considered as the second reinforcement for the manufacturing of hybrid aluminum metal matrix composites (HAMMCs) due to their extraordinary mechanical and tribological, electrical, and thermal properties [6,22,23]. The amount of Si_3_N_4_ was kept constant at 5 wt.%, whereas the amount of GNPs was varied from 0.5 to 1.5 wt.% to study their effect on the structural, and mechanical properties of the synthesized HAMMCs. It is observed that the amount of GNPs has a substantial impact on the structural and mechanical properties of the HAMMCs. The novelty of the present work resides in the fact that the composites were fabricated through a microwave sintering approach, which is quite efficient and provides cost-effectiveness, better process control, meticulous microstructure, improved properties and scaling up in the process. The development of Al-Si_3_N_4_-GNPs through a microwave sintering approach has not been reported earlier.

## 2. Materials and Methods

Pure Al powder (99.5% purity, Alfa Aesar, Tewksbury, MA, USA) with a particle size of 7–15 µm was used as the matrix material. Crystalline Si_3_N_4_ powder (98.5+% purity, Alfa Aesar, Tewksbury, MA, USA) with a particle size of 15–30 nm and GNPs powder (97% purity, Cheap Tubes, Cambridge port, VT, USA) with an average particle size of 8–15 nm were used as reinforcement materials for the synthesis of Al-Si_3_N_4_-GNPs composites. The Si_3_N_4_ contents were fixed while the amount of GNPs was varied in the Al matrix to develop Al-Si_3_N_4_-GNPs composites. The composition of fabricated Al-Si_3_N_4_-GNPs composites is presented in Table 1.

The powder metallurgy method was employed for the fabrication of the Al-Si_3_N_4_-GNPs composites utilizing ball milling and microwave sintering techniques. Powders of the matrix and reinforcements were weighed using an analytical balance (Sartorius, ENTRIS64-1S, Lower Saxony, Germany) and then were uniformly mixed using a planetary ball mill (PM200, RETSCH 20.640.0001, Haan, Germany). Powders were blended at room temperature for a duration of 2 h at a milling speed set at 200 RPM to obtain a homogenous distribution of the nanoparticle reinforcements in the matrix. The blended powder mixtures (~1.0 g) were cold-compressed under an applied pressure of 50 MPa and a residence time of 1 min to form cylindrical pellets. The green pellets were sintered using a bi-directional microwave-assisted rapid sintering technique supplied by (VB ceramic furnace, VBCC/MF/1600 °C/14/15, Chennai, India). The operating conditions of the microwave sintering were programmed at a temperature of 550 °C utilizing a rapid heating rate of 10°C /min and a dwell time of 30 min. After microwave sintering, the sintered pellets were cooled down to room temperature (25 °C) and then characterized using various techniques. An illustration of the experimental procedure for the fabrication of Al-Si_3_N_4_-GNPs composites is presented in Figure 1.

Phase analysis of microwave sintered hybrid composites was carried out using an X-ray diffractometer (XRD) (PANalytical X’pert pro, PANalytical B.V., Almelo, The Netherlands). The XRD spectra were documented in the range of 2θ = 20–80° at 45 kV and 40 mA employing Cu anode radiation, with a scanning rate of 1.5°/min and a step size of 0.02°. Microstructural analysis was performed on the polished sintered pellets using a field emission scanning electron microscope (FE-SEM) (SEM-FEI Nova NanoSEM 450 FE-SEM, Hillsboro, OR, USA). Elemental mapping analysis was conducted with Energy-dispersion X-ray spectroscopy (EDS) (Bruker SDD-EDS, Coventry, UK) to evaluate the composition and phase purity of the fabricated HAMMCs.

The density and porosity of the Al-Si_3_N_4_-GNPs composites were calculated based on Archimedes’ principle. Density kit analytical balance (Sartorius YDK03, Lower Saxony, Germany) with a precision of ± 0.0001 g was used for the calculation. The surface topography of the fabricated Al-Si_3_N_4_-GNPs composites was examined using an atomic force microscope (AFM) (MFP-3D AFM, Asylum Research, Oxford, UK). Nano-hardness and Young’s modulus were analyzed at room temperature conditions using a nanoindenter (MFP-3D Nano Indenter, Asylum Research, Oxford, UK). Forces were applied to the samples at a maximum load of 1 mN, and dwell time of 5 s at the peak load. The indentation displacement was observed and documented within the nm range.

The Microhardness of the developed Al-Si_3_N_4_-GNPs composites was examined using a Vickers tester (FM-ARS9000, MKV-h21, Tokyo, Japan). Five successive iterations were undertaken on each specimen to achieve high accuracy in results with an applied load of 5 gf and a dwell time of 10 s. Compression analysis of the composites was accomplished using a universal testing machine (Lloyd, USA-LR50Kplus, Sussex, UK) at standard conditions of temperature and pressure under an engineering strain rate of 0.6 mm/min. The documented data was an average value of four successive test results. The fractographic analysis was conducted on the fractured surfaces of Al-Si_3_N_4_-GNPs composites using a field emission scanning electron microscope (SEM-FEI Nova NanoSEM 450 FE-SEM, OR, USA) to understand the mode of fracture during the compression testing.

## 3. Results and Discussion

### 3.1. Characterization of Raw Materials

Figure 2 represents the FE-SEM images of (a) pure Al, (b) Si_3_N_4_ and (c) GNPs powders utilized in the fabrication of Al-Si_3_N_4_-GNPs composites. The morphology of the pure Al and Si_3_N_4_ can be clearly seen in Figure 2a,b, respectively, while Figure 2c depicts the plate-like structure of GNPs. The provided micrographs confirm the size of the reinforcements in the nanometric. The scale bar on FE-SEM images presented in Figure 2b,c clearly depicts the nanometric size of the reinforcements confirming the development of composites.

### 3.2. XRD Analysis of Al-Si_3_N_4_-GNPs Composites

Figure 3a illustrates the XRD spectra of the microwave-sintered pure Al and the manufactured Al-Si_3_N_4_-GNPs composites consisting of disparate weight fractions of GNPs. The results verify the presence of Si_3_N_4_ and GNPs in the Al matrix. It is noticed that the intensity of the Al peaks is high due to its high concentration in the Al-Si_3_N_4_-GNPs composites indicating its crystalline character. However, the low intensity of peaks of the reinforcements (Si_3_N_4_ and GNPs) are detected due to their low amount in the Al-Si_3_N_4_-GNP composites, which is consistent with the previous studies [24]. Moreover, XRD results did not show any traces of undesired phases or impurities such as aluminum carbide (Al_4_C_3_) for S3, S4 and S5 due to the low contents of GNPs incorporated in the Al matrix [25,26]. It can also be observed that the intensity of the GNPs reinforcement peaks rises with the increase in the concentration of the reinforcement. The enlarged section in Figure 3a from the diffraction angle 20–35° demonstrates the augmentation of the intensity of GNPs with the increase of the reinforcement content in the Al matrix. XRD patterns of S5 were magnified and illustrated in Figure 3b to clearly signify the presence of Al, crystalline Si_3_N_4_, and GNPs in the matrix. Additionally, it confirms the absence of other impurities in the Al-Si_3_N_4_-GNPs composites showing close adherence with the previous studies [27,28]. The XRD results indicate that phase pure Al-Si_3_N_4_-GNPs composites containing various concentrations of GNPs is successfully developed through the microwave-assisted powder metallurgy route.

### 3.3. Microstructural Analysis of Al-Si_3_N_4_-GNPs Composites

Figure 4 represents FE-SEM images of the fabricated Al-Si_3_N_4_-GNPs composites containing various weight fractions of GNPs. The FE-SEM images reveal the distribution of Si3N4 (white areas indicated with blue arrows) and GNPs (light grey areas indicated with green arrows) are evenly distributed within the Al matrix (dark grey area). Homogenous distribution of Si_3_N_4_ nanoparticles in the Al matrix was observed in the sample S2 as shown in Figure 4b. On the other hand, some agglomeration of reinforcement is also detected in some areas of samples S3, S4, and S5. The agglomeration tendency increased with the increase of GNPs in Al-Si_3_N_4_-GNPs composites as shown in Figure 4c–e. The agglomeration of particles has also been previously reported and can be attributed to the electrostatic attractive forces [24,29].

Figure 5 represents the EDX analysis of the fabricated Al-Si_3_N_4_-GNPs composites (S3, S4, and S5) as depicted in Figure 5a–c with their corresponding elemental mapping images (Figure 5d–f) showing elemental distribution. The images indicate the presence of Al, Si, N, and C as the main elements in the composites. The composition of S3, S4, and S5 is also presented in Table 2. It is worth mentioning that the illustrated elemental mapping images have proven the uniform and even distribution of the reinforcements in the fabricated Al-Si_3_N_4_-GNPs composites with a small agglomeration.

### 3.4. Density and Porosity of Al-Si_3_N_4_-GNPs Composites

Table 3 demonstrates the experimental outcomes for the measured density and porosity of the developed Al-Si_3_N_4_-GNP composites by varying GNP contents in the Al matrix. It can be noticed that the measured density of the Al matrix has increased with the addition of the crystalline Si_3_N_4_ nanoparticles in the matrix (S2) owing to its high theoretical density Si_3_N_4_ (3.17 g/cm^3^) in comparison to that of the Al base metal (2.7 g/cm^3^) [30]. The decline in the behavior of the density is observed with the inclusion of GNPs to the Al matrix which could be justified due to the low theoretical density of GNPs (2.267 g/cm^3^) [9]. This is so because the contents of Si_3_N_4_ are fixed in the composites and the incorporation of GNPs in the matrix is thus accomplished at the replacement of Al contents. Furthermore, the density of the composites is found to decrease with the increasing amount of GNPs due to an incremental decrease in the density of the Al matrix due to its replaced amount by GNPs. A similar trend in the density results was noticed in a previous study conducted on Al-Graphene composites [10,17]. For a deeper insight into the density behavior of the composites, relative density ($\rho_{r}$) of the developed composites was determined using the actual or measured density ($\rho_{c}$) and the theoretical density values ($\rho_{t}$) as indicated in the below Equation [31].
(1)$$\rho_{r}\left(\%\right)\;=\;(\rho_{c}/\rho_{t})\times \;100$$

Table 3 shows a higher relative density of Al-Si_3_N_4_ composite (S2) (~95.95%) than that of Al matrix (S1) (~95.91%), which indicates the decent interfacial interaction at the Al-Si_3_N_4_ interface. However, a decrease in the relative density has been noticed with the increase of the GNP content in the Al matrix, where it reached its terminal value at (~94.22%). This decrease in the densities with the addition of GNPs is in line with the already published reports [6].

Concurrently, the percentage of porosity was calculated and found to be decreasing after the addition of Si_3_N_4_ nanoparticles to the Al matrix (S2), as shown in Table 3. The good dispersion and uniform distribution of Si_3_N_4_ within the Al matrix has resulted in a lack of agglomerations of the reinforcement and thus reduced the porosity of the Al-Si_3_N_4_ composite [32]. However, with the successive addition of GNPs to the composites, the porosity level is noticed to increase due to agglomeration of GNPs in the Al-Si_3_N_4_-GNPs composites (S3, S4, and S5) as presented in Table 3.These findings are consistent with previous studies [33].

### 3.5. Surface Topography Al-Si_3_N_4_-GNPs Composites

The surface topography of the fabricated composites was examined using atomic force microscopy. Figure 6 represents the AFM images of synthesized Al-Si_3_N_4_-GNP composites. The surface roughness was evaluated using the root-mean-square (RMS) roughness parameter which described the smoothness of the surface. The measured RMS value of the pure Al matrix is (16.37 nm). The surface roughness of the composites is noticed to increase with the addition of Si_3_N_4_ used as a reinforcement (34.7 nm). There is a further increase in the RMS value by the addition of GNPs (49.11 nm). The RMS value rises with the rising quantity of GNPs in the Al-Si_3_N_4_-GNPs composites and it reaches its maximum value at (65.32 nm) at the highest compositions (Al-5Si_3_N_4_-1.5GNPs). This behavior of incremental increase in the surface roughness of Al-Si_3_N_4_-GNPs composites could be ascribed to the amalgamation of hard ceramic nanoparticles into the Al matrix and the formation of agglomerated Si_3_N_4_ and GNPs within the Al matrix as observed in FE-SEM images Figure 4c–e. A similar trend of roughness increment was noticed in the previous studies [34,35].

### 3.6. Nanoindentation Analysis and Microhardness of Al-Si_3_N_4_-G NPs Composites

The load-indentation depth curves of the pure Al and developed Al-Si_3_N_4_-GNP composites containing different contents of GNPs are presented in Figure 7a. It can be noticed that the indentation depth decreases with the increasing content of reinforcements, suggesting an improvement in the hardness of the Al-Si_3_N_4_-GNPs composites in comparison to the pure Al. For a clear comparison, the hardness and Young’s modulus of pure aluminum (S1) and the various developed Al-Si_3_N_4_-GNP composites are also presented in Figure 7b. It can be observed that the hardness and Young’s modulus of Al-Si_3_N_4_-GNPs composites increase with increasing concentration of GNPs. Such behavior is attributed to the presence of GNPs having a positive influence in improving the mechanical properties of pure Al [6,17]. Besides, the presence of hard crystalline Si_3_N_4_ nanoparticles has also contributed to improving the hardness of the pure Al matrix [36]. The highest value of hardness is demonstrated by sample S5 (0.899 ± 45.5 GPa) contributing an enhancement of ~32.1% as compared to sample S1 (pure aluminum, 0.68 ± 52.65 GPa). Moreover, Young’s modulus of the developed composite was found to improve from (6.23 ± 0.45 GPa) at S1 in pure aluminum to (10.55 ± 1.00 GPa) at S5 in the highest GNPs content of the composites, offering ~69.3% of development as compared to pure Al.

A comparison of microhardness of pure Al and developed Al-Si_3_N_4_-GNPs composites is also illustrated in Figure 7c. It can be noticed that the microhardness values of the developed Al-Si_3_N_4_-GNPs composites have exhibited better performance than that of the pure Al matrix (64.03 ± 1.86 HV). As indicated in the graph, the microhardness is gradually increasing in conjunction with the addition of GNPs contents in the Al matrix. The terminal value of microhardness (86.72 ± 0.55 HV) is obtained in sample S5, which contains maximum contents of GNPs contributing an improvement of ~35.4%. An explanation for the rise of microhardness with the addition of reinforcement from its initial value could be attributed to the dispersion hardening effect as a result of the presence of hard reinforcements [6] and the uniform distribution of Si_3_N_4_ and GNPs avoiding excessive agglomeration. More importantly, the phenomenon of dispersion hardening has played a critical role in increasing the hardness of the pure Al matrix [14]. Additionally, the enhancement in hardness with the increasing amount of reinforcement can also be explained by the rule of the mixture as stated below [37].
(2)$$H_{c}=H_{m}F_{m}\;+\;H_{r}F_{r}$$where $H_{c}$, $H_{m}$ and $H_{r}$ represent the hardness of the composite, matrix and reinforcement, respectively, and $F_{m}$ and $F_{r}$ represent the volume fraction of the matrix and reinforcement, respectively.

### 3.7. Compression Analysis of Al-Si_3_N_4_-GNPs Composites

Figure 8 represents the compression test results of the developed Al-Si_3_N_4_-GNPs composites with various content of GNPs. Figure 8a represents the engineering stress/strain curve of the developed Al-Si_3_N_4_-GNPs composites. The corresponding compressive yield strength (CYS) and ultimate compression strength (UCS) of the composites are illustrated in Figure 8b. It can be noticed that the compression strength has increased significantly with the addition of Si_3_N_4_ particles to the Al matrix in (S2) where it has reached its highest UCS value at (357 ± 5 MPa) and at 0.2% offset CYS value of (103 ± 4 MPa). This behavior of improvement in mechanical properties is a consequence of the effective load transfer $\sigma_{load}$ which occurs between the ductile Al matrix and the hard-ceramic phase (S_i3_N_4_) reinforcement. The interfacial interaction between the reinforcement phase and the matrix promotes the load transfer and is presented by the following equation [38].
(3)$$\sigma_{load}=0.5V_{f}\sigma_{YM}$$

The volume fraction of the ceramic reinforcement in the equation is represented by $V_{f}$ and $\sigma_{YM}$ denotes the yield stress of the matrix.

Moreover, upon the crack propagation, the resistant to dislocation offered by the reinforcement (secondary phase) significantly increase the strength of the Al-Si_3_N_4_-GNPs composites, which is in agreement with Orowan mechanism [39]. Additionally, the interaction of the dislocation and secondary phase particles creates a dislocation loop that resists the propagation of crack [36], uniformly dispersed hard reinforcement (Si_3_N_4_) throughout the matrix promotes the nucleation of dislocation loops [40,41]. Furthermore, the work of dispersion hardening supports the strength of the material to withstand the maximum applied load [42,43].

The ultimate compressive strength has diminished with the addition of the GNPs to Al-Si_3_N_4_-GNPs composites where it reached its terminal value at (312 ± 7 MPa) and at 0.2% offset CYS value of (72 ± 2 MPa) at a uniform failure strain of 0.63 in S5 sample. This drop in the compressive strength can be attributed to the agglomeration of GNPs in the composites due to excessive amount, hence weakens the interface between Al particles and GNPs leading to strength deterioration [24,29,44]. Furthermore, the grain boundary spaces or pores increases with the increasing content of GNPs because of the pi–pi interaction between different graphene nanoplatelets [45]. The porosity in Al-Si_3_N_4_-GNPs composites has a negative effect on the sinterability of these composites. The presence of pores due to GNPs agglomerates or overlapping of GNPs leads to crack initiation during the compressive loading and hence adversely affects the compressive strength of the developed composites. Another reason is due to the lubricating nature of graphene which triggers the graphene sheets to slide easily under compression loading, hence, it reduces the friction between hard ceramic particles of Si_3_N_4_ allowing it to slide during the plastic deformation region under compressive loading which leads to a degradation in the compressive strength [17]. Additionally, the weak Van der Waals force between the graphene sheets enhances the shifting of the sheets under applied loading which in turns decreases the compressive strength [46].

However, even with this drop of strength, the CYS of developed Al-Si_3_N_4_-GNPs composites (S3, S4, and S5) is exhibiting improvement of ~87.7%, ~65.3% and ~47% respectively when compared to the pure Al matrix. Moreover, the UCS of developed Al-Si_3_N_4_-GNPs composites (S3, S4, and S5) have also shown an improvement of ~12.4%, ~8.1%, and ~2% respectively when compared to the pure Al matrix. These findings are consistent with the previous studies [24,47]. These improved values of CYS and UCS of (S3, S4, and S5) have exhibited the beneficial role of the inclusion of GNP reinforcement into the Al matrix.

**Figure 8 materials-14-01898-f008:**
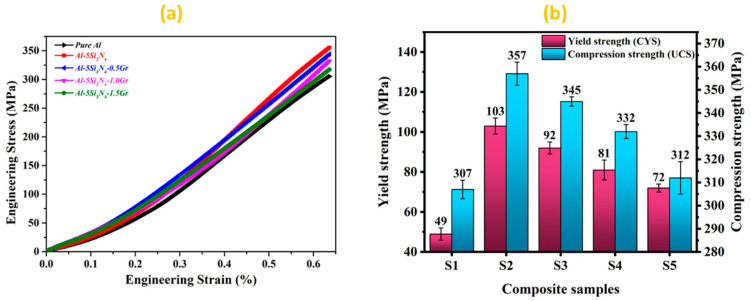
(**a**) Engineering stress-strain curve under compressive loading (**b**) Compressive yield strength (CYS) and ultimate compressive strength (UCS) values of Al-Si_3_N_4_-GNPs composite samples.

### 3.8. Fractography of Al-Si_3_N_4_-GNPs Composites

Figure 9 shows SEM representations of the fractured surfaces of Al and Al-Si_3_N_4_-GNPs composites under compression loadings. The provided images demonstrate the shear mode fracture that occurred in the composites along with cracks on the surfaces of the samples in the direction of the compressive loading axis at a 45° angle. As indicated from the images, the compressive deformation of the Al used as a matrix is different from that of Al-Si_3_N_4_-GNPs composites, due to the work hardening effect and the heterogeneous deformation [14,19].

## 4. Conclusions

In this study, Al-Si_3_N_4_-GNPs composites containing different concentrations of GNPs were successfully synthesized using the microwave-assisted powder metallurgy method. The structural (XRD) and compositional analyses (EDX) confirm the formation of phase pure Al-Si_3_N_4_-GNPs composites having an agglomeration effect with increasing concentration of GNPs. The density of the prepared composites decreases with the increasing amount of GNPs, while the porosity follows an opposite trend. The surface roughness of the Al-Si_3_N_4_-GNPs composites increases with the exhibit promising hardness as compared to pure Al. Although, the values of CYS and UCS of Al-Si_3_N_4_-GNPs composites decrease with the increasing amount of GNPs but remain higher than the pure Al justifying the motivation of their development. A shear mode of fracture is prevalent in Al-Si_3_N_4_-GNPs composites under compressive loading.

## Figures and Tables

**Figure 1 materials-14-01898-f001:**
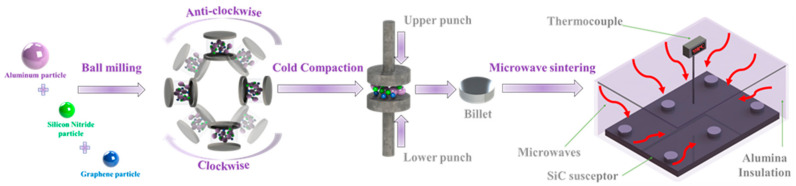
Schematic representation for the development of Al-Si_3_N_4_-GNPs composites.

**Figure 2 materials-14-01898-f002:**
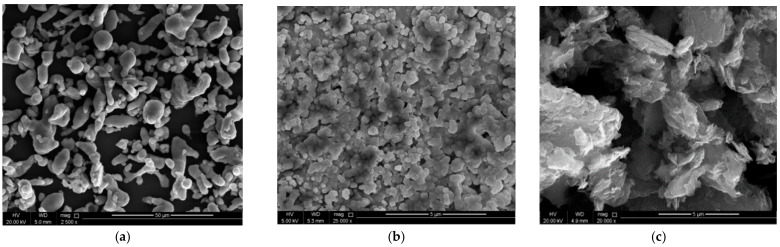
FE-SEM images of as received powders of (**a**) Pure Al, (**b**) Si_3_N_4_ and (**c**) GNPs.

**Figure 3 materials-14-01898-f003:**
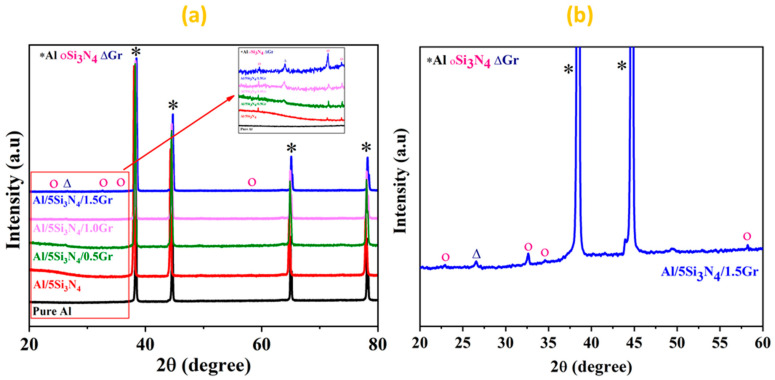
X-ray diffraction patterns of (**a**) Al-Si_3_N_4_-GNPs composites (inset graph shows the enlarged section that covers the 2θ range 20–35°) and (**b**) Magnified pattern of Al-5Si_3_N_4_-1.5GNPs composites.

**Figure 4 materials-14-01898-f004:**
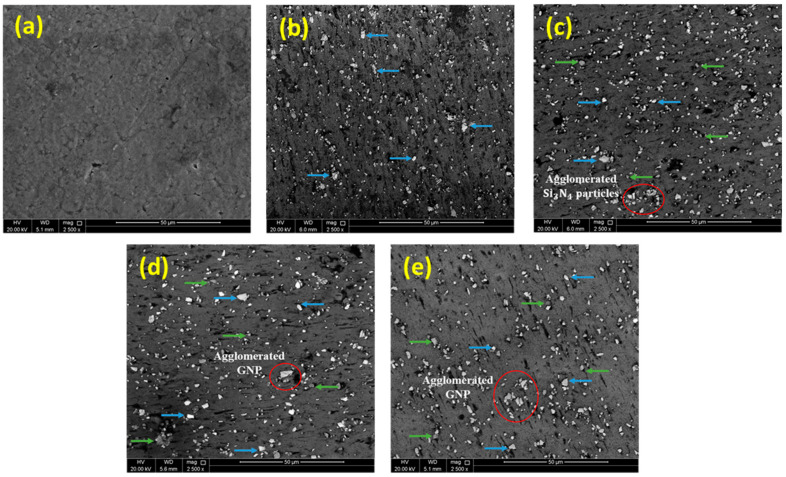
FE-SEM images of (**a**) Pure Al, (**b**) Al-5Si_3_N_4_, (**c**) Al-5Si_3_N_4_-0.5GNPs, (**d**) Al-5Si_3_N_4_-1GNPs and (**e**) Al-5Si_3_N_4_-1.5GNPs composites.

**Figure 5 materials-14-01898-f005:**
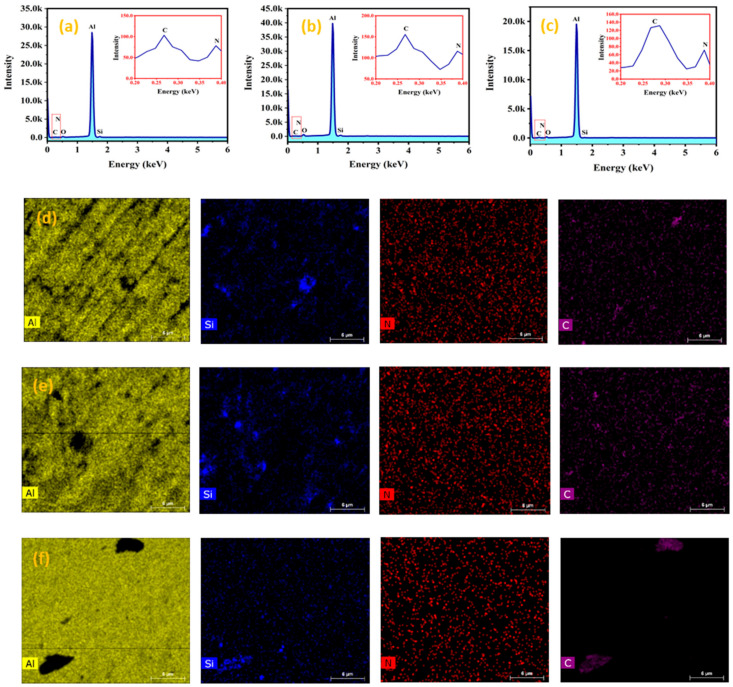
(**a**–**c**) Energy dispersive X-ray spectroscopy spectrum analysis and (**d**–**f**) elemental mapping images of Al-5Si_3_N_4_-0.5GNPs, Al-5Si_3_N_4_-1GNPs and Al-5Si_3_N_4_-1.5GNPs composites (scale-6 µm).

**Figure 6 materials-14-01898-f006:**
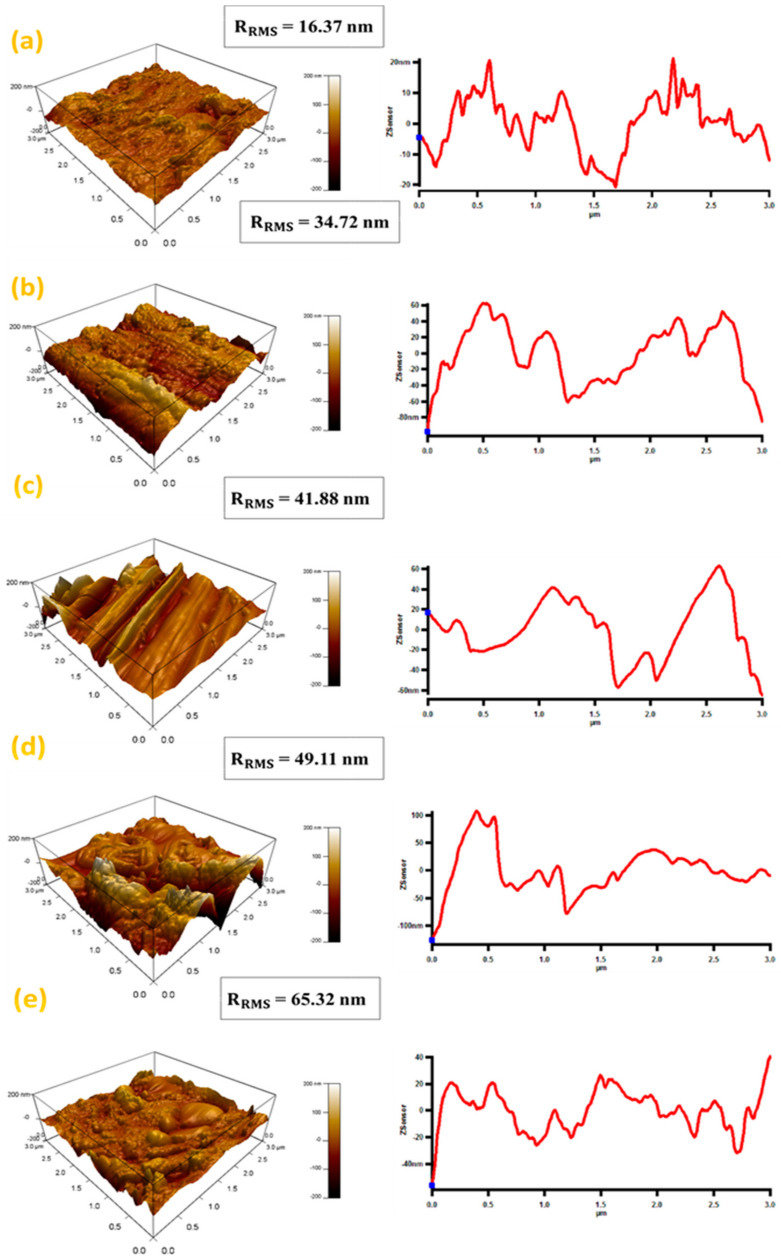
2D and 3D atomic force microscope images of (**a**) Pure Al, (**b**) Al-5Si_3_N_4_, (**c**) Al-5Si_3_N_4_-0.5GNPs, (**d**) Al-5Si_3_N_4_-1GNPs and (**e**) Al-5Si_3_N_4_-1.5GNPs composites.

**Figure 7 materials-14-01898-f007:**
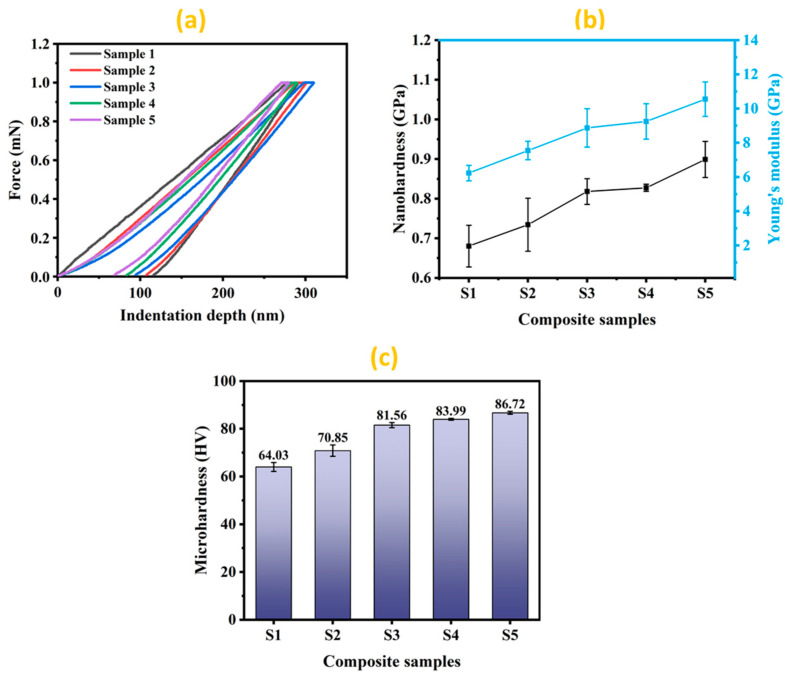
(**a**) Load-indentation depth curves, (**b**) Nanohardness and Young’s modulus and (**c**) Microhardness values of Al-Si_3_N_4_-GNPs composites.

**Figure 9 materials-14-01898-f009:**
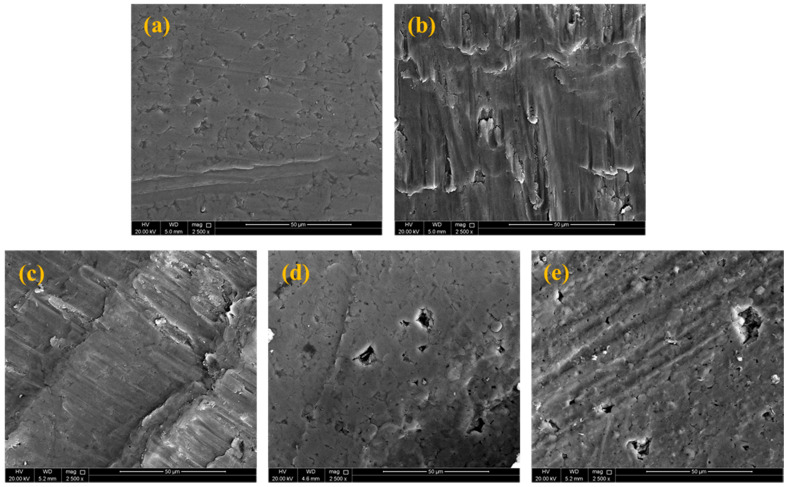
Compression fracture images of (**a**) Pure Al, (**b**) Al-5Si_3_N_4_, (**c**) Al-5Si_3_N_4_-0.5GNPs, (**d**) Al-5Si_3_N_4_-1GNPs and (**e**) Al-5Si_3_N_4_-1.5GNPs composites.

**Table 1 materials-14-01898-t001:** Composition of Al-Si_3_N_4_-GNPs composites.

Sample No.	Sample Title	Weight Fraction (wt.%)
1	S1	Pure Al
2	S2	Al-5 wt.% Si_3_N_4_
3	S3	Al-5 wt.% Si_3_N_4_-0.5 wt.% GNPs
4	S4	Al-5 wt.% Si_3_N_4_-1.0 wt.% GNPs
5	S5	Al-5 wt.% Si_3_N_4_-1.5 wt.% GNPs

**Table 2 materials-14-01898-t002:** Elemental distribution of Al-5Si_3_N_4_-0.5GNPs, Al-5Si_3_N_4_-1GNPs and Al-5Si_3_N_4_-1.5GNP composites in correspondence to EDX spectrum Figure 5a–c.

Element	Weight Percent (wt.%)		
	S3	S4	S5
Al	74.42	68.35	65.25
Si	1.65	1.59	0.22
N	5.30	5.94	3.43
C	9.74	10.99	22.31
O	8.89	13.13	8.79
Total	100	100	100

**Table 3 materials-14-01898-t003:** Density and porosity measurements of Al-Si_3_N_4_-GNPs composites.

Composite Samples	Theoretical Density (g/cm^3^)	Experimental Density (g/ cm^3^)	Relative Density (%)	Porosity (%)
S1	2.700	2.568 ± 0.002	95.10	4.90 ± 0.058
S2	2.720	2.610 ± 0.003	95.95	4.05 ± 0.100
S3	2.718	2.576 ± 0.002	94.81	5.20 ± 0.084
S4	2.715	2.560 ± 0.006	94.30	5.70 ± 0.239
S5	2.712	2.556 ± 0.006	94.22	5.78 ± 0.231

## Data Availability

Not applicable.

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
