# Peer review of "Effect of Silicon Nitride and Graphene Nanoplatelets on the Properties of Aluminum Metal Matrix Composites"

_materials, 2021, doi:10.3390/ma14081898_

Round 1
Reviewer 1 Report
The main drawback of the manuscript, which hamper recommending it for publication, is as follows. The research methods used in this study do not allow revealing the nanoscale structural elements of the material investigated explicitly or indirectly. The authors have not shown reliably that the metal matrix composite is nanostructured.
There are several mistakes that can be fixed easily:
- in line 122, when describing the X-ray diffraction pattern in Figure 2 (b), for some unknown reason, the authors indicate that the phases Si3N4 and GNPs are detected in the amorphous state. Whereas on the pattern, the corresponding peaks from the crystalline phases are visible;
- the extremely high carbon content in the material of composites, given in table 2, is doubtful. Doubts are confirmed by visualization on maps of the distribution of elements in Figure 4;
- in line 155 the sentence is not completed;
- after line 155 there is a fragment of table 2;
- it is doubtful that reference [11] is correct;
- there is no reference [32] in the text.
Author Response
Kindly find the attached file including the responses to the comments

Reviewer 2 Report
The scientific research paper “Effect of silicon nitride and graphene nanoplatelets on the properties of aluminum metal matrix nanocomposites” - 1140850 ”by Rokaya Abdelatty , Adnan Khan , Moinuddin Yusuf , Abdullah Alashraf and Rana Abdul Shakoor is a successful attempt for the Center for Advanced Materials (CAM), Qatar University Doha in the field of aluminum metal matrix nanocomposites.
The work is very concise. The authors included in the paper what they proposed and studied. The introduction has a multitude of references (22) which covers the analyzed domain of nano sized reinforcements in AMMCs. Different types of reinforcements such as metallic, amorphous, and ceramic particles, have been utilized in AMMCs for enhancing their performances. The article proposes the solution with Si3N4-GNPs reinforcements for aluminum. The materials chosen for the research are of very good quality. Powders were blended at room temperature for a duration of 2 hrs at a milling speed set at 200 RPM to obtain a homogeneous distribution of the nanoparticle reinforcements in the matrix. Schematic diagram for the development of Al-Si3N4-GNPs nanocomposites is accurately shown. The density and porosity of the Al-Si3N4-GNPs nanocomposites were calculated by conducting based on Archime des’ principle. Materials, methods and apparatus have been selected and presented with great accuracy, and the research equipment (USA, Germany, Holland, UK, Japan) is very judiciously chosen and is top.
XRD analysis of Al-Si3N4-GNPs nanocomposites are consistent with other specialized studies and indicate the presence of Al, amorphous Si3N4, and GNPs in the matrix. In the microstructural analysis of Al-Si3N4-GNPs nanocomposites are accurately indicated (by arrows) the distribution of Si3N4 and GNPs uniformly distributed within the Al matrix. Elemental distribution of Al-5Si3N4-0.5GNPs, Al-5Si3N4-1GNPs and Al-5Si3N4-1.5GNPs nanocomposites in correspondence to EDX spectrum. The aluminum content decreases as the reinforcement content increases.
The density of the prepared nanocomposites decreases logically with the increasing amount of GNPs, and the porosity shows an opposite trend.
The surface roughness of Al-Si3N4-GNPs nanocomposites is dependent on the hardness of the composite material. I also think that more reinforcement compositions should have been needed, but the paper allows other researchers to clarify the use of these reinforcements in aluminum.
In the abstract and in also, in the paper is a nonconcordance: ”This research work aims at investigating the influence of contents of silicon nitride (Si3N4) and graphene nanoplatelets (GNPs) on the physical (density, structural, morphological) and mechanical properties (microhardness, nanoindentation) of Al-Si3N4-GNPs nanocomposites”. The scientific paper addresses only the influence of the amount of the Si3N4 contents were fixed while the amount of GNPs was varied in the Al matrix to develop Al-Si3N4-GNPs nanocomposites.
Author Response

(The authors gave the same response as above.)

Reviewer 3 Report
The work is aimed at the study of influence of contents of silicon nitride (Si3N4) and graphene nanoplatelets on the physical and mechanical properties of Al-Si3N4-8 GNPs nanocomposites.
Although the paper may be of some interest, there are a lot of shortcomings.
- A great amount of papers has been written on this similar subject. It would be very useful if the authors clearly specify the novelty of the work and the state beyond the state of the art in a field.
- As silicon nitride (Si3N4) GNP are widely studied as reinforcement materials, please explain the motivation and the purpose and the expected improvement in properties of the AMMC.
- It is not clear where and how the material under investigation may be of practical interest.
- Why was the amount of Si3N4 chosen as 5 wt. % (why not less or higher), and the amount of GNPs was varied from 0.5 to 1.5 wt.%?
- Intro: Metal matrix composites consist of not just and only lightweight metallic matrix, but any metallic matrix (depending on the purpose). Please, re-phrase.
- Intro: As “the reinforcement material is required to be uniformly distributed within the matrix”, please describe the methods and/or approaches to achieve this uniform distribution, especially in terms of GNPs.
- It is stated that “Fig. 2 shows the XRD spectra of the microwave sintered pure Al and the manufactured Al-Si3N4-GNPs nanocomposites containing different weight fractions of GNPs”; however, Fig. 2 gives “X-ray diffraction (XRD) patterns of (a) Al-Si3N4-GNPs nanocomposites and (b) Magnified pattern of Al-5wt.% Si3N4-1.5wt.% GNPs nanocomposites”. Please, specify clearly what is shown in Fig.1. Fig 2b could be an insert in Fig 1a. The designation of phases is not obviously demonstrated. Please improve the quality of Fig. 2.
- The XRD does not show any traces of undesired phases or impurities. It is stated that it is due to low sensitivity of the device. Does this mean that, in principal, it is very probable to find other phases with another device?
- Actually, from Fig 2, it is impossible to make a statement of increasing intensity with concentration (although it is well-understood, the particular XRD pattern does not obviously confirm this)
- The discussion on density is obvious. As densities of all constituents are known, it would be more interesting to learn about relative density as the measured density gives no additional information on performance in this case.
- It is not completely clear why with the addition of GNPs to the nanocomposites, the porosity level increases. Why does agglomeration of GNPs result in increase in porosity?
- Study on surface topography, although being interesting, seems to be out of scope of the paper. Please clarify why such kind of studies are important. What additional useful information may be obtained from this study?
- An increase in hardness in composite (as compared to pure Al) is an obvious finding. It is not clear why and what for the nanoindentation tests were performed.
- In part of “discussion”, there is no actual discussion, just reporting the results, which are well-predicted. Please discuss the mechanism of hardening in composites.
- What was the reason to perform a compressive test, not a tensile one?
- It is stated that “Si3N4 reinforcement particles in the Al matrix which provides a higher resistivity to the plastic deformation region and prevents the dislocation movement in the matrix”. That is a well-established fact explained at any textbook. Please confirm that in the case of the composites developed, the reinforcement particles indeed prevent the dislocation motion.
- Please explain/discuss on the statements that “drop in the compressive strength can be attributed to the agglomeration of GNPs” and “the lubricating nature of graphene which triggers the graphene sheets to slide easily under compression loading”.
- Please discuss on the friction between Si3N4 particles, which in your statement, results in degradation of the compressive strength.
Author Response

(The authors gave the same response as above.)

Round 2
Reviewer 1 Report
The authors have done large work to improve the manuscript, taking into account most of the comments and recommendations of the reviewers. Among other things, the authors showed that the initial components of the composite were nanostructured materials. However, it does not follow from this that the composite obtained by sintering is also a nanostructured material. Until it is clearly demonstrated that the composite contains nanoscale structural elements, it cannot be called a nanocomposite.
I recommend excluding mentioning of the nanoscale structure of the composite from the title and text of the manuscript.
Author Response
The authors are thankful to the reviewer for his valuable suggestion. The word "nanocomposite" has been replaced by "composite" in the title and text of the revised manuscript.
Reviewer 3 Report
The papers is much improved.
English should be carefully checked.
Author Response
We are thankful for the appreciation of the reviewer. The revised manuscript has been thoroughly checked for grammatical mistakes. A few grammatical corrections have been highlighted in the revised version of the manuscript.